# Vitamin C and Benzoic Acid Intake in Patients with Kidney Disease: Is There Risk of Benzene Exposure?

**DOI:** 10.3390/nu18010132

**Published:** 2025-12-31

**Authors:** Manuela Yepes-Calderón, Caecilia S. E. Doorenbos, Eva Corpeleijn, Casper F. M. Franssen, Michel J. Vos, Daan J. Touw, Christophe Mariat, Annelies E. de Weerd, Stephan J. L. Bakker

**Affiliations:** 1Division of Nephrology, Department of Internal Medicine, University of Groningen, University Medical Center Groningen, 9700 RB Groningen, The Netherlandss.j.l.bakker@umcg.nl (S.J.L.B.); 2Department of Epidemiology, University of Groningen, University Medical Center Groningen, 9700 RB Groningen, The Netherlands; 3Department of Laboratory Medicine, University of Groningen, University Medical Center Groningen, 9700 RB Groningen, The Netherlands; 4Department of Pharmacology, University of Groningen, University Medical Center Groningen, 9700 RB Groningen, The Netherlands; 5Division of Nephrology, Department of Internal Medicine, Hôpital Nord CHU de Sainte-Etienne, 42270 Saint-Etienne, France; 6Division of Nephrology, Department of Internal Medicine, Erasmus Medical Centre, 3015 GD Rotterdam, The Netherlands

**Keywords:** vitamin C, benzoic acid, benzoate, benzene, chronic kidney disease

## Abstract

Vitamin C is a small water-soluble molecule primarily cleared by the kidneys. Therefore, its plasma concentration would be expected to increase as kidney function declines. However, studies in patients with chronic kidney disease (CKD) and kidney transplant recipients have shown the opposite: a positive correlation between kidney function and plasma vitamin C levels. In this review, we discuss potential explanations for this counterintuitive finding and suggest alternative mechanisms influencing vitamin C bioavailability in this population. We also explore the hypothesis that this phenomenon may be linked to benzoic acid (benzoate) exposure. Benzoic acid is a widely used food preservative that, like vitamin C, is water-soluble and renally excreted. In individuals with impaired kidney function, reduced clearance may lead to elevated circulating benzoic acid levels, which could increase the likelihood of an in vivo chemical reaction between benzoic acid and vitamin C, resulting in the formation of benzene, which is a known toxic and carcinogenic compound. We summarize experimental evidence demonstrating the vitamin C–benzoic acid reaction in vitro, along with preliminary animal studies suggesting it may also occur in vivo. We also discuss the potential clinical consequences of benzene exposure in the context of patients with kidney function impairment. Given the widespread use of benzoic acid as a food preservative and the ongoing discussion around vitamin C supplementation in patients with kidney disease, this review invites further investigation to evaluate whether this reaction represents a health hazard for this population.

## 1. Introduction and Methods

Vitamin C is water-soluble and renally cleared, so levels would be expected to rise as kidney function declines. Yet, in patients with chronic kidney disease (CKD) and kidney transplant recipients (KTR), studies report the opposite: a positive correlation between kidney function and plasma vitamin C independent of intake. This review interrogates that paradox by evaluating alternative mechanisms influencing vitamin C metabolism in these populations and by exploring a linked hypothesis that benzoate exposure, a common food preservative, may interact with vitamin C in vivo to generate benzene, with potential clinical consequences in patients with impaired kidney function.

To achieve this, we conducted a narrative review focusing on these areas: vitamin C pharmacokinetics, vitamin C levels in patients with kidney disease, reasons for vitamin C deficiency in this group, sources and exposure to benzoate in the general population, and benzene formation both in vitro, as well as the likelihood of its formation in vivo. We searched MEDLINE/PubMed, Embase, Cochrane Library, and Web of Science (Jan 1990–Nov 2025; English), as well as guidelines and regulatory repositories (NIH ODS, EFSA, FDA). Strategies combined controlled vocabulary and free text for ascorbic acid, benzoic acid/benzoate, hippurate, benzene, and CKD/KTR. Eligible evidence included randomized trials, cohort studies, case–control and cross-sectional studies, mechanistic/occurrence studies relevant to in vivo plausibility, and authoritative guidelines. The synthesis of evidence was qualitative. Chat GPT version 5 was used for the purposes of minor grammar improvements throughout the manuscript.

## 2. Vitamin C

Vitamin C (ascorbic acid) is a water-soluble molecule with two important biological roles: enzyme cofactor and antioxidant [1] (Figure 1). As a cofactor, vitamin C is a one-electron donor that reduces the active sites of iron- and copper-dependent hydroxylases [2], enabling, among others, collagen maturation (prolyl/lysyl hydroxylases) [3], carnitine biosynthesis [4], and catecholamine production (dopamine β-hydroxylase) [5]. These functions are needed for wound healing, vascular integrity, energy metabolism, and neurotransmission [1]. As an antioxidant, vitamin C regenerates vitamin E and stabilizes reactive nitrogen/oxygen species [6], supports epithelial barriers and leukocyte function [7], and enhances non-heme iron absorption by reducing Fe^3+^ to Fe^2+^ in the gut [8]. Given the relevance of its biological tasks, the maintenance of adequate vitamin C levels is a tightly regulated biological function.

The pharmacokinetics of vitamin C is summarized in Figure 2. As humans lack the mechanisms to synthesize vitamin C from glucose and galactose like many other species (e.g., higher primates, guinea pigs, and bats), vitamin C is an essential nutrient for them [1]. The primary natural sources of vitamin C are fresh fruit and vegetables, while there is also a growing availability of supplements in both oral and intravenous form [9]. For adults, the typical recommended daily amount (RDA) ranges from 75 to 110 mg/day, depending on sex [9,10]. In practice, this can be easily achieved through diet, as fruits are rich sources of this nutrient. For example, a medium orange provides roughly 70 mg, a medium kiwifruit contains about 64–71 mg, and half a cup of raw red bell pepper has approximately 95 mg [9]. After oral intake, absorption occurs in the small intestine in a nonlinear manner via the epithelial sodium-dependent vitamin C transporters (SVCTs) [11]. The fractional absorption via this transporter stays high up to doses of around 200 mg and then declines as transporters approach saturation. At the 200 mg oral doses, the bioavailability of vitamin C is around 100% but can decrease to less than 50% on larger oral doses (around 1250 mg) [11]. Importantly, the bioavailability of vitamin C from food is equal to that from supplementation [12]. Normal plasma concentration ranges from ~60 to 80 µM [13]. In the case of intravenous dosing, vitamin C bypasses intestinal control and can transiently raise plasma into the millimolar range, roughly 30–70× higher than oral, before clearance and re-establishment of the normal plasma concentration [14].

The further distribution of vitamin C is highly compartmentalized and depends on the tissue-specific expression levels and subtypes of the SVCTs, resulting in a diverse range of organ concentrations of vitamin C at homeostasis, ranging from about 0.2 mM in skeletal muscles and heart to up to 10 mM in the brain and adrenal glands [11]. As a highly hydrophilic low-molecular-weight compound, excretion occurs mainly via the kidney. Vitamin C is freely filtered at the glomerulus. Afterwards, the proximal tubule avidly reabsorbs filtered ascorbate across the apical membrane via SVCT1 transporters [11,14]. This reclamation is saturable and, therefore, nonlinear. In healthy adults, it is reported that little-to-no urinary loss occurs up to doses of ~100 mg, and that the excreted fraction sharply increases after oral doses ≥ 500 mg [13].

Definitions of vitamin C inadequacy and deficiency vary across the literature. Plasma vitamin C levels of ≥50 µmol/L appear to offer optimal protection against cardiovascular- and cancer-related morbidity and mortality, according to evidence from multiple studies and regulatory agencies [10,15,16,17]. Therefore, levels < 50 µmol/L can be considered suboptimal and are, by some studies, classified as inadequate [18]. More widely accepted definitions of vitamin C inadequacy vary from levels < 23–28 µmol/L and are linked to increased risk for subclinical symptoms and health concerns, whereas levels < 10–12 µmol/L are widely accepted as a definition of clinical vitamin C deficiency, with severely increased risk of adverse health outcomes, including scurvy [16,19,20,21].

## 3. Vitamin C Status in Patients with Chronic Kidney Disease

As glomerular filtration rate (GFR) declines, one might expect higher plasma vitamin C levels due to reduced renal clearance. Paradoxically, prior research shows that individuals with lower GFR exhibit lower plasma vitamin C levels.

Low plasma vitamin C levels are observed across the entire spectrum of chronic kidney disease (CKD). In non-dialysis CKD (average eGFR 23 mL/min/1.73 m^2^), average vitamin C concentrations of 35 µmol/L have been reported [22]. In a recent study, 80% of patients with non-dialysis stage 4–5 CKD had inadequate levels (<35 µmol/L), with 17% being deficient (≤10 µM) [23]. This finding was supported by another study involving patients with stage 4–5 CKD, both with and without dialysis, where 79% were found to be deficient (vitamin C < 11 µmol/L) [24]. Older studies also indicated that patients with kidney failure, whether on hemodialysis or not, exhibited lower plasma vitamin C levels compared to matched healthy controls [25].

In dialysis-dependent CKD, vitamin C concentrations are further reduced. Studies report that the average pre-dialysis vitamin C levels in hemodialysis patients are around 12 µmol/L, compared to 27 µmol/L in non-dialysis CKD stages 3–5 patients, with levels decreasing further to, on average, 7 µmol/L after dialysis treatment [26]. Similarly, another study found that 16 of 19 dialysis patients had inadequate pre-dialysis concentrations (<23 µmol/L), which decreased by an additional 33% during dialysis [27], while a large multicenter study reported deficiency (<11 µmol/L) in 36% of patients across different dialysis modalities [28]. In peritoneal dialysis, concentrations < 30 µmol/L were observed in 40% of patients [29].

After kidney transplantation, vitamin C levels remain low. Yepes-Calderon (2024) reported inadequate vitamin C concentrations (≤28 µmol/L) in 46% of kidney transplant recipients (KTR) at 6 months post-transplant and in 30% at 12 months [30]. In another large KTR cohort, 22% had inadequate levels (≤28 µmol/L) at a median of 5.9 years post-transplant, with deficiency independently associated with increased mortality after correction for, e.g., time since transplantation and graft function [31].

### 3.1. Mechanisms of Vitamin C Deficiency in CKD

There are several potential mechanisms underlying the widely observed vitamin C deficiency among patients with CKD.

#### 3.1.1. Inadequate Dietary Intake

Vitamin C intake among patients with CKD is often inadequate [23,24]. Also, patients with serious CKD often have a loss of appetite [32] and often show malnutrition [33,34], which may contribute to low vitamin C intake. In this population, the dietary potassium restriction may also further contribute to diets low in vitamin C, as many foods containing vitamin C, such as fruits and vegetables, are high in potassium, while potassium tends to be avoided because of fear of hyperkalemia [19].

#### 3.1.2. Increases Consumption

Among people with CKD, persistent low-grade inflammation and oxidative stress, which are hallmarks of the uremic milieu, can accelerate turnover of antioxidant nutrients such as vitamin C [35,36]. Several uremia-related solutes further deplete vitamin C. Uremic toxins like indoxyl sulfate, *p*-cresyl sulfate, and hippurate can amplify reactive oxygen species and redox cycling [37,38]. Others, like myeloperoxidase-derived hypochlorous acid and lipid aldehydes, can oxidize ascorbate directly [36,39,40]. Dicarbonyls such as methylglyoxal and glyoxal can also form covalent adducts with ascorbate [41]. Together, these processes convert ascorbate to dehydroascorbic acid with subsequent degradation, or sequester it in adducts, which lowers measurable plasma vitamin C relative to intake.

This mechanism has also been proposed as a reason for vitamin C deficiency in KTR [42,43]. Consistent with this, in a cohort comparing diabetic and non-diabetic CKD, the inverse association between vitamin C and eGFR was stronger among patients with diabetes, who also showed tighter links between inflammatory biomarkers and vitamin C levels [22]. Moreover, despite average intakes approximating the general population’s RDA, KTR displayed a higher prevalence of vitamin C deficiency consistent with increased requirements in this group [44,45,46,47,48].

#### 3.1.3. Impaired Tubular Reabsorption

As previously described, vitamin C homeostasis depends on renal tubular reabsorption mediated by SVCT1/2 [11]. When plasma vitamin C concentrations exceed the absorptive capacity, which Ebenuwa et al. estimated to be 48.5 ± 5.2 µM for healthy men and 58.3 ± 7.5 µM for healthy women [49], excess vitamin C is excreted in urine. Impaired reabsorption can lead to inappropriate urinary losses, despite reduced GFR and despite inadequate plasma concentrations, which has recently been shown in patients with diabetes [49] and Fabry disease [50].

#### 3.1.4. Medication

Some medications commonly used in CKD care also lower plasma vitamin C or increase its loss, like acetylsalicylic acid (aspirin), loop diuretics, and proton pump inhibitors (PPIs). Aspirin can impair gastrointestinal absorption and increase urinary excretion, loop diuretics raise urinary losses, and PPIs reduce plasma and gastric vitamin C (via higher gastric pH) [1,2,3,4,5,6,7,8].

#### 3.1.5. Removal by Dialysis

Substantial vitamin C loss during dialysis has been recognized since the inception of the therapy [51]. Hemodialysis sessions commonly reduce plasma vitamin C by ~33–75% [26,52,53,54]. The amount recovered in dialysate often exceeds the estimated extracellular pool (fractional removal > 1), implying mobilization from intracellular compartments [23]. Losses are larger with hemodialysis than with peritoneal dialysis, and plasma levels are correspondingly lower in hemodialysis patients [23,28]. In the literature, daily vitamin C losses of 58 (27–127) mg per session were observed in conventional hemodialysis compared to 128 (97–156) mg in nocturnal hemodialysis [23] and average daily absolute losses of 45 mg in nine patients on peritoneal dialysis [29]. Greater vitamin C losses in hemodialysis may result from more efficient removal of water-soluble substances compared to peritoneal dialysis [55,56].

#### 3.1.6. Benzoic Acid Exposure and Benzene Production

Finally, we hypothesize that a potential mechanism could be benzene formation in situ through an ascorbic acid–benzoic acid pathway. In the rest of this review, we summarize evidence on human benzoic acid exposure, outline the biochemical mechanism of benzene formation, and critically evaluate the evidence regarding its plausibility in vivo (Figure 3).

## 4. Benzoic Acid: Exposure, Pharmacokinetics, and Renal Excretion

Benzoic acid (C6H5COOH), benzoate (C6H5COO^−^), and benzoate salts (E210–E213) are widely used antimicrobial preservatives, primarily in acidic foods and drinks such as soft drinks, fruit juices, and pickles [57,58,59,60]. They can also be found as a natural constituent of vegetal and animal products, especially dairy [58,61]. The goods manufacturing practices of the US Food and Drug Administration recommend that the content in food as a preservative should not exceed 0.1% [57,60]. On average, according to the last World Health Organization (WHO) monitoring report (2021), the concentrations benzoic acid do not exceed 40 mg/kg of food for naturally occurring benzoic acid and 2000 mg/kg of food for added benzoic acid [58]. The acceptable daily intake (ADI) for benzoate preservatives varies per authority. While the European Food and Safety Authority re-evaluated its limit in 2016 and settled on a maximum of 5 mg/kg body weight per day (bw/day) of benzoic acid only [60], the Food and Agriculture Organization of the United Nations and the WHO agreed to an ADI of 0–20 mg/kg bw (kilogram of bodyweight)/day in 2021 for benzoic acid, benzoate salts, and related benzyl derivatives [59]. Average intake varies greatly per nation and is also dependent on individual eating patterns. In 2000, the WHO reported that the national mean intake of benzoic acid ranged from 0.18 mg/kg bw/day (Japan) to 2.3 mg/kg bw/day (USA). However, groups of “high consumers” were identified, in which the average benzoic acid consumption could reach from 7.3 mg/kg bw/day (USA) to 14 mg/kg bw/day (China) [58]. In the 2021 update, benzoic acid intake ranged from 0.07 to 3.21 mg/kg bw/day in the general population in Europe and, in the “high exposure group,”, it could reach as high as 7.1 mg/kg bw/day [59,60]. Consistently across countries, the food category contributing most to benzoic acid intake was soft drinks [57,58].

After ingestion, benzoic acid is rapidly and extensively absorbed from the gastrointestinal tract and transported to the liver, where it is activated to benzoyl-CoA and conjugated with glycine to form hippuric acid [62]. This is a near-complete (~75–100%) conversion with plasma peaks of hippuric acid around 1–2 h post-dose [62,63]. It is important to note that this mechanism can be saturated with really high doses, as demonstrated by early studies with mega boluses of benzoic acid of 160 mg/Kg [64]. After conjugation, hepatocytes export it across their basolateral (sinusoidal) membrane into the hepatic venous blood, from which it circulates systemically and is excreted via the kidneys. Renal handling of hippuric acid involves both filtration allowed by its molecular weight of 0.179 kDa and active tubular secretion, and this occurs rapidly [64,65,66]. Around 75–100% of an oral dose of benzoic acid is recovered in urine as hippuric acid within 6–24 h, with the small remainder excreted over the next 1–2 days, and no accumulation is expected under typical dietary exposure [67]. Because virtually all benzoic acid is metabolized and eliminated via this route, the maximum urinary excretion rate of hippuric acid following administration of sodium benzoate is considered to be near the highest rate of metabolism [64].

Given that the hepatic conjugation is considered the rate-limiting step in benzoic acid metabolism, few studies have evaluated the effects of impaired kidney function in benzoic acid/hippuric acid elimination. A first study exploring whether hepatic benzoic acid metabolism was affected in patients with kidney failure found that benzoic acid conjugation remains largely intact in this population. [68] A second study showed that, as expected, the overall clearance of hippuric acid was inversely related to kidney function. This study also demonstrated that adaptive mechanisms occur in the context of advanced kidney disease, with an apparent increase in both glomerular filtration and tubular excretion in residual nephrons [67]. However, the cleared fraction of hippuric acid still remained smaller in individuals with kidney disease than in healthy controls [69].

## 5. Benzene Formation and Biological Effects

Under certain circumstances, benzoic acid can react to form benzene. Benzene formation from benzoic acid occurs through a decarboxylation reaction, where the carboxyl group is removed as carbon dioxide, yielding benzene (Figure 3). Under laboratory conditions, this process is facilitated by strong heating with bases (such as sodium hydroxide with calcium oxide), reductive decarboxylation (using reducing agents like zinc or iron), or pyrolysis at high temperatures.

In food and beverages, the decarboxylation of benzoic acid to benzene is mediated primarily by free radicals. This radical-mediated pathway is significantly accelerated by the simultaneous presence of vitamin C, transition metal ions (such as Cu(II) or Fe(III)), and heat, acidic conditions, or light exposure. Vitamin C reduces molecular oxygen (catalyzed by metal ions), generating hydrogen peroxide and, ultimately, hydroxyl radicals. These hydroxyl radicals can then attack benzoic acid, extracting an electron, forming a benzoic acid radical which loses CO_2_ and yields benzene [70,71]. Heat and light accelerate radical generation (e.g., via riboflavin photo-oxidation), whereas metal chelators and some antioxidants suppress it. In beverages, measurable benzene appears at parts-per-billion levels when benzoic acid and ascorbate are together, especially with elevated temperature, low pH, or UV exposure, so manufacturers control metal ions, pH, and storage conditions to minimize formation [71,72,73]. Vitamin C’s role in this process is both catalytic, enabling radical generation, and sacrificial, as it is oxidized to dehydroascorbic acid (DHA) in the process that leads to benzene production. This DHA in cells can later be reduced back to ascorbate [70].

Although this chemistry is widely documented in vitro and in foods [74], it is uncertain whether this or a similar reaction can occur in vivo. In liquid models, formation of benzene is strongly pH-dependent and favored under acidic conditions. Several studies and guidance documents report maximal formation around pH ≈ 2, a steep drop from pH 3 to 5, and no detectable benzene at ~pH 7 [70]. Still, the ingredients and catalysts for this reaction exist in biological milieus: benzoic acid can be ingested or formed metabolically; ascorbic acid circulates at micromolar levels; and redox-active metals plus Fenton chemistry can generate hydroxyl radicals [75]. Acidic microenvironments inside the human body where such chemistry could, in principle, be favored include the stomach (fasted pH ~1–2; fed ~3–5) and the endolysosomal compartments (pH ~4.5–5.5) [76,77]. An animal pharmacological study from the mid-20th century demonstrated that the administration of approximately 0.02 mg/kg of (halo)benzoic acids to rats resulted in a marked depletion of adrenal ascorbic acid within hours. This reduction was originally interpreted as a secondary effect of adrenal cortical activation and increased ACTH secretion, since vitamin C serves as a cofactor in corticosteroid biosynthesis. However, such findings may also suggest that benzoic acid and related compounds can enhance ascorbate turnover within tissues [78,79]. These findings do not prove in-body benzene generation, but they suggest a pro-oxidant context in which radical decarboxylation is chemically plausible.

The potential extent of benzene exposure related to this mechanism in humans cannot be calculated. Although average dietary benzoic acid intake, the extent of conjugation to hippuric acid, and plasma vitamin C concentrations have been reported in the literature, key determinants remain unknown, including the plasma concentration and residence time of free benzoate, reaction kinetics under physiological conditions, the influence of the local redox environment, and clearance mechanisms, particularly in patients with impaired kidney function. As a result, any numerical estimation would rely on speculative assumptions, highlighting a critical gap in current human data.

Benzene exposure has multiple well-documented adverse effects on human health and is classified as a Group 1 human carcinogenic. Benzene primarily affects the hematopoietic system but also has documented effects on other organs and systems such as the central nervous system and mucosal surfaces [74,80,81]. Some of the adverse effects after benzene exposure include anemia, leukopenia, thrombocytopenia, acute myelogenous leukemia, myelodysplastic syndrome, and immune disfunction [82]. A recent study showed that even long-term exposure to low concentrations was associated with increased risk of all-cause mortality, cardiovascular disease, cancer, and respiratory disease [83].

## 6. Knowledge Gaps and Future Perspectives

Multiple uncertainties remain around the magnitude of exposure, the mechanistic plausibility, and the clinical relevance of concurrent vitamin C and benzoic acid exposure with potential in situ benzene formation. Clarifying these issues is increasingly important given the widespread use of benzoic acid in processed foods and beverages [60] and the discussion of whether vitamin C supplementation should be considered in patients with kidney disease given their higher rate of deficiency [23].

Uncertainty about benzoic acid exposure magnitude arises because current assessments rarely integrate all sources simultaneously (foods, beverages, medicines such as therapeutic sodium benzoate, and personal care products) or capture co-ingestion patterns (e.g., acidic beverages formulated with benzoic acid consumed alongside vitamin C) that could modulate ascorbate–benzoic acid chemistry [60,84,85]. In addition, genetic and pharmacologic modifiers of key transporters and enzymes—SVCT1/2 for vitamin C [86], OAT1/3 for hippurate and related organic anions [87], and CYP2E1 for benzene metabolism—remain to be characterized and incorporated into individual-level exposure and risk models.

Regarding the plausibility of in vivo benzene formation, human evidence is limited as most research has been conducted in vitro in product matrices such as beverages or focuses on biomarkers for environmental or occupational exposure. [84,85]. There are no controlled human studies in which subjects were simultaneously exposed to ascorbic and benzoic acid and in which benzene formation was monitored or demonstrated in vivo. Furthermore, the few human investigations rarely employ specific benzene biomarkers, such as trans,trans-muconic acid (ttMA) or urinary S-phenylmercapturic acid (S-PMA) [88]. Both ttMA and S-PMA are established metabolites formed via hepatic benzene oxidation: ttMA arises from the ring opening of benzene oxide through muconaldehyde intermediates, while S-PMA results from the conjugation of benzene oxide with glutathione [89]. Of these, S-PMA is considered more specific at low exposure levels. Nevertheless, ttMA remains a practical and widely used biomarker for population-level benzene exposure due to its higher urinary concentrations and simpler analytical requirements [88,89]. The confirmation of whether this reaction is possible in vivo is especially important for vulnerable populations such as patients with reduced kidney function, where the risk of vitamin C deficiency is higher, among others, because of their higher expenditure [35,36], and where potential benzoic acid exposure might also be higher due to their decreased hippurate clearance [69,90]. However, there is, so far, no consistent evidence that benzoic acid or benzene levels are higher in the population with CKD.

We aim to partially address these knowledge gaps through the forthcoming Benzoic Acid and Nutritional and Clinical Health in Kidney Transplantation (BANCH) study. This crossover randomized clinical trial will assign kidney transplant recipients to either continue their usual diet or follow a two-week benzoic acid-free diet [91]. Plasma vitamin C concentrations and urinary ttMA excretion will be assessed at the end of each intervention period. Participant recruitment is scheduled to begin in 2026, and the findings are expected to provide further insight into the potential for in vivo benzene formation. We consider that clarifying the risk of benzene exposure in patients with decreased kidney function is critical, particularly given the growing CKD population and the wide-spread use of benzoic acid as a food preservative [60].

## Figures and Tables

**Figure 1 nutrients-18-00132-f001:**
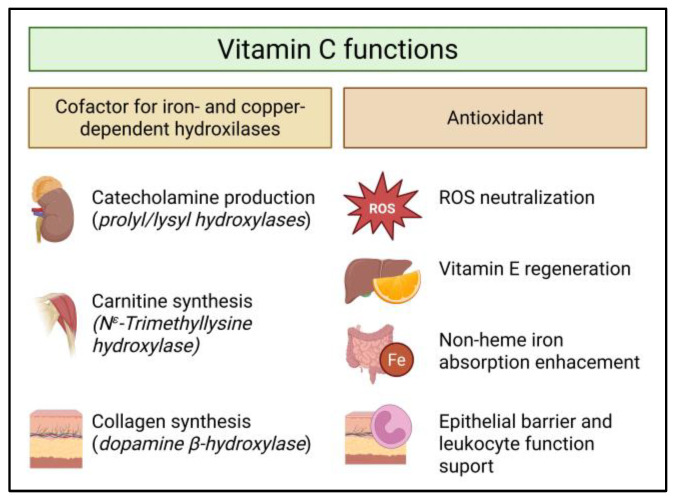
Vitamin C functions in humans. ROS: reactive oxygen species.

**Figure 2 nutrients-18-00132-f002:**
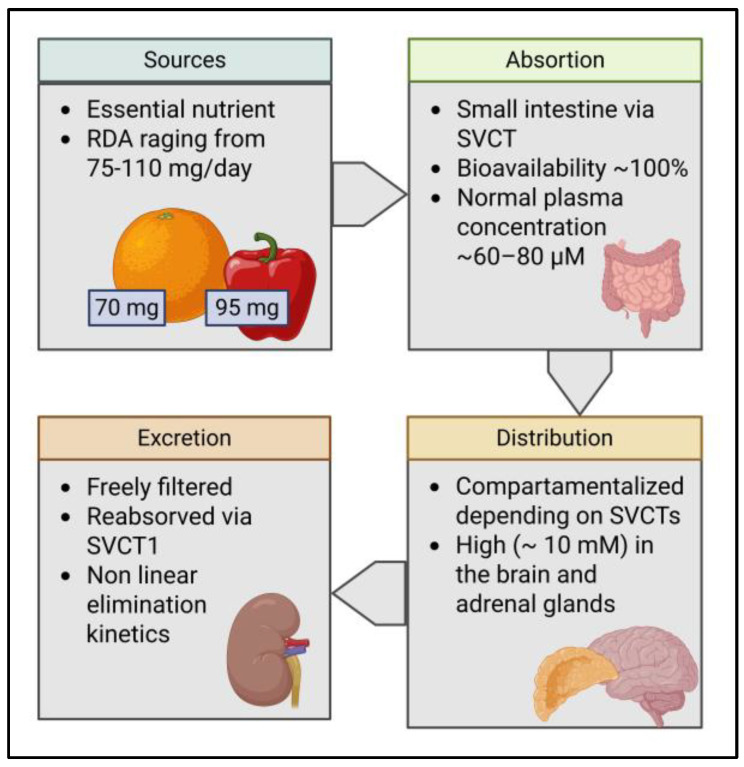
Pharmacokinetics of Vitamin C. RDA, recommended daily amount; SVCT, sodium-dependent vitamin C transporter.

**Figure 3 nutrients-18-00132-f003:**
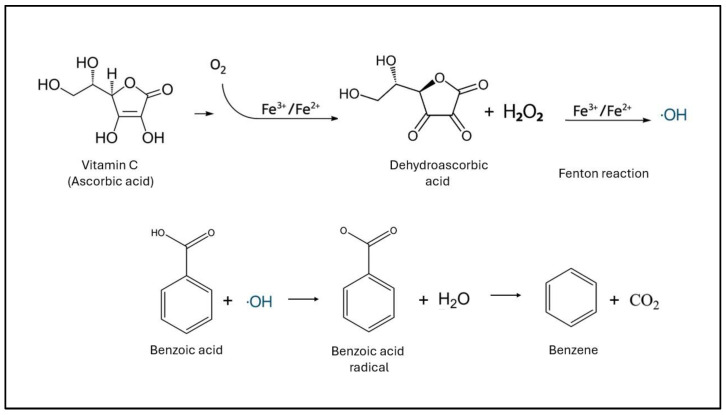
Benzene formation.

## Data Availability

No new data were created or analyzed in this study. Data sharing is not applicable to this article.

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
