# Peer review of "Vitamin C and Benzoic Acid Intake in Patients with Kidney Disease: Is There Risk of Benzene Exposure?"

_nutrients, 2025, doi:10.3390/nu18010132_

Round 1
Reviewer 1 Report
Comments and Suggestions for Authors
This review is very well written, on an intriguing and timely topic, with significant implications for public health.
I would like the authors to comment in the final chapter on these considerations that seem to emerge from the literature proposed in the review:
1) There are no controlled human studies in which healthy subjects drank beverages containing benzoate (or benzoic acid) + ascorbic acid and had specific biomarkers of benzene exposure (t,t-MA, S-PMA, or U-benzene) measured before and after with a design demonstrating significant in-vivo formation. Existing publications focus primarily on: (1) benzene formation in the beverage (during production/storage) and (2) validation of biomarkers for environmental or occupational exposures.
2) There are some studies reporting the presence and variation of benzene (or volatile compounds that include it) in the blood/expired air of patients with end-stage renal disease or during hemodialysis, but there is no large and consistent literature that definitively demonstrates that tissue or plasma levels of benzene are systematically higher in patients with CKD, on hemodialysis or renal transplant recipients compared to the general population.
3) Additionally, the BANCH project—“Benzoic Acid and Nutritional and Clinical Health in Kidney Transplantation”—is underway. This project is described in an official document as an ERA grant for a one-center fellowship at the University Medical Center Groningen (UMCG), with PI Bakker. I believe this should be properly explained in the review and merits a summary of the preliminary results, which I believe support the hypotheses suggested in the review.
Reviewer 2 Report
Comments and Suggestions for Authors
General Comments
This review discusses why plasma vitamin C is low in patients with chronic kidney disease and kidney transplant recipients, despite reduced renal clearance. It summarizes known causes (diet, inflammation, dialysis, etc.) and introduces a hypothesis that concurrent exposure to benzoic acid/benzoate and vitamin C may lead to in vivo benzene formation. However, the study has several limitations that should be addressed before publication.
Comments:
- In the author's hypothesis that concurrent vitamin C and benzoic acid exposure could lead to in vivo benzene formation, it would be better if the authors could provide more quantitative context. Specifically, based on typical dietary benzoate intake, its near-complete conjugation to hippuric acid, and the reported plasma vitamin C ranges in CKD/KTR. A simple back-of-the-envelope calculation or schematic would also help assess the magnitude of potential in vivo risk.
- The author mentions that human evidence on in vivo benzene formation from vitamin C-benzoate interaction is sparse and that future studies should use specific biomarkers such as ttMA and S-PMA. This would be essential information, and the author could expand this part by summarizing existing human biomonitoring studies in which dietary benzoate and vitamin C intake were associated with benzene biomarkers.
- The logical bridge from hippurate accumulation and oxidative stress in CKD to increased benzene formation is not fully convincing, because chemistry is primarily discussed for benzoic acid plus ascorbate. It would also be better if the author could clearly distinguish between well-established findings and speculation.
Round 2
Reviewer 1 Report
Comments and Suggestions for Authors
I have no further comments